# Single-Nucleotide Polymorphisms in the Thioredoxin Antioxidant System and Their Association with Diabetic Nephropathy in Slovenian Patients with Type 2 Diabetes—A Preliminary Study

**DOI:** 10.3390/ijms26051832

**Published:** 2025-02-20

**Authors:** Jernej Letonja, Petra Nussdorfer, Danijel Petrovič

**Affiliations:** 1Laboratory for Histology and Genetics of Atherosclerosis and Microvascular Diseases, Institute of Histology and Embryology, Faculty of Medicine, University of Ljubljana, Korytkova 2, 1000 Ljubljana, Slovenia; jernej.letonja@mf.uni-lj.si (J.L.); petra.nussdorfer@mf.uni-lj.si (P.N.); 2Institute of Histology and Embryology, Faculty of Medicine, University of Ljubljana, Vrazov trg 2, 1000 Ljubljana, Slovenia

**Keywords:** diabetic nephropathy, T2DM, thioredoxin 2, thioredoxin reductase 2, TXNIP, polymorphism, rs814011, rs1548357, rs7212

## Abstract

Diabetic nephropathy (DN) is a microvascular complication of type 2 diabetes mellitus (T2DM) that develops after years of T2DM and affects approximately one in four diabetic patients. Thioredoxin (TXN), thioredoxin reductase (TXNRD), and thioredoxin-interacting protein (TXNIP) are part of the thioredoxin antioxidant system, which is involved in DN. We included 897 Slovenian patients with T2DM lasting more than 10 years in our preliminary study. In total, 344 patients with DN were included in our case group, while 553 without DN comprised our control group. The genotypes of *TXN2* rs8140110, *TXNRD2* rs1548357, and *TXNIP* rs7212 were determined for all participants using real-time PCR. We found a statistically significant association between the T allele of the *TXN2* rs8140110 polymorphism and DN (*p* < 0.001; OR: 0.52; 95% CI: 0.36–0.74). The TT and TC genotypes were also significantly less likely to develop DN in comparison to the CC genotype according to the dominant model of inheritance (*p* < 0.001; OR: 0.51; 95 CI: 0.34–0.75). We did not find a statistically significant association between rs1548357 or rs7212 and DN. To conclude, the rs8140110 polymorphism in the TXN2 gene is associated with DN in Slovenian patients with T2DM.

## 1. Introduction

Type 2 diabetes mellitus (T2DM) is a chronic multisystemic disease that affected an estimated 536.6 million people worldwide in 2021. The prevalence is increasing, and it is estimated that more than 780 million people will suffer from diabetes in 2045 [1]. Complications related to diabetes can be classified as microvascular or macrovascular and they severely affect quality of life and are related to increased mortality [2]. Diabetic nephropathy (DN) is a microvascular complication of T2DM that develops after years of T2DM and affects approximately one in four diabetic patients [3]. DN is characterized by a gradual worsening of kidney function and is the most common cause of chronic kidney disease (CKD) and end-stage kidney disease (ESKD) [4].

The pathogenesis of DN is multifactorial and complex; however, it is still not fully understood. Chronic hyperglycemia initiates the development of DN by inducing oxidative stress in endothelial cells. The increased production of reactive oxygen species (ROS) decreases the antioxidant capacity and promotes the release of inflammatory cytokines that drive inflammation [5]. Mitochondria are crucial in ROS production since the majority of ROS are produced in the mitochondrial respiratory chain [6]. Because of this, the mitochondria are well-equipped with various antioxidant defenses such as the glutathione antioxidant system, thioredoxin antioxidant system, catalase, and superoxide dismutase. ROS are a crucial part of normal physiological processes, but they are also critical in the pathogenesis of DN [5,6].

The thioredoxin antioxidant system is a thiol-dependent antioxidant system that is also involved in the repair of oxidized proteins. It consists of thioredoxin (Txn), thioredoxin reductase (TxnRD), NADPH, and thioredoxin-interacting protein (TXNIP). There are different thioredoxin antioxidant systems based on their location in the cell. Txn1 and TxnRD1 are located in the cytosol; and Txn2, TxnRD2, and TxnRD3 are located in the mitochondria [7,8].

The Txn–TXNIP interaction is an important regulator of apoptosis. Txn1 and Txn2 directly bind with apoptosis signal-regulating kinase 1 (ASK1) and inhibit its activity. During states of oxidative stress, TXNIP is transferred to the mitochondria where it causes the release of ASK1 which leads to apoptosis [9]. The thioredoxin system is also involved in inflammation. Mitochondrial ROS induces inflammation by NLRP3 inflammasome activation. TXNIP induces inflammation in oxidative stress through its interaction with the NLRP3 inflammasome which causes the release of interleukin-1 β (IL-1β) and interleukin-18 (IL-18) [9]. The Forkhead box class O1 (FOXO1) is also involved in the regulation of Txn–TXNIP in oxidative stress in the kidney [10]. TXNIP levels in the kidney tissues, serum, and urine are significantly higher in patients with T2DM DKD than in healthy controls [11].

The TXNRD2 gene is associated with glaucoma [12,13], prostate cancer [14], dilated cardiomyopathy [15], and isolated glucocorticoid deficiency [16]. The gene for Txn2 is located on 22q12.3, and rs8140110 is located in the intronic region of the gene. T is the ancestral allele and its frequency is 11% in the European population. The gene for TxnRD2 is located on 22q11.21, and rs1548357 is located in the intronic region of the gene. C is the ancestral allele and its frequency is 30% in the European population. The gene for TXNIP is on 1q21.1, and rs7212 is located in the 3’ region. G is the ancestral allele, and the frequency of C which is the minor allele is 4.5% in the European population [17].

The aim of our preliminary study was to determine whether there is an association between the selected polymorphisms of the thioredoxin antioxidant system and DN in Slovenian patients with T2DM.

## 2. Results

The clinical and laboratory characteristics of the participants are presented in Table 1. There was no statistically significant difference between the control group (participants with T2DM without DN) and the case group (participants with T2DM and DN) in age, sex, diastolic blood pressure, BMI, smoking status, presence of CVD, hemoglobin, total cholesterol, HDL-cholesterol, or LDL-cholesterol. The participants in the case group had a longer duration of T2DM (*p* < 0.001) and hypertension (*p* = 0.0058), a higher systolic blood pressure (*p* < 0.001), a larger waist circumference (*p* < 0.001), a higher prevalence of diabetic neuropathy (*p* < 0.001) and DR (*p* < 0.001), and higher levels of glucose (*p* < 0.001), HbA1c (*p* < 0.001), cystatin C (*p* < 0.001), triglycerides (*p* = 0.0123), and albumin–creatinine ratio in all three samples (*p* < 0.001) in comparison with the control group. The participants in the case group had a shorter duration of DR (*p* < 0.001) and a smaller eGFR (*p* < 0.001) than the control group.

The distribution of rs8140110 alleles and genotypes is presented in Table 2. There was a statistically significant difference in the distribution of both alleles and genotypes between the case and control groups. There was a lower frequency of the TT and TC genotypes in the case group in comparison to the control group (*p* = 0.0322). The T allele was less frequent in the case group in comparison to the control group (*p* = 0.0093). The distribution of genotypes did not significantly differ from the Hardy–Weinberg equilibrium. The distribution of genotypes was also significantly different according to the dominant model of inheritance (*p* = 0.0238), but not the recessive one. We used logistic regression analysis to assess the investigation between rs8140110 and DN after adjusting for the waist duration of type 2 diabetes mellitus, duration of hypertension, systolic blood pressure, waist circumference, diabetic retinopathy, diabetic neuropathy, HbA1c, fasting glucose, urea, creatinine, cystatin C, and urine albumin–creatinine ratio. The association was still statistically significant for the TT vs. CC comparison (*p* = 0.0498), TC vs. CC (*p* = 0.0034), the allele distribution (*p* < 0.001), and in the dominant model of inheritance (TT + TC vs. CC) (*p* < 0.001).

The distribution of TXNRD2 rs1548357 and TXNIP rs7212 alleles and genotypes as well as the dominant and recessive models of inheritance and the results of logistic regression are presented in Table 3 and Table 4, respectively. There was no statistically significant difference in the distribution of rs1548357 or rs7212 alleles or genotypes between the case and control groups. The distribution of genotypes did not significantly differ from the Hardy–Weinberg equilibrium.

The minor allele frequency (MAF) of rs8140110 was 8% and 11.8% in the case and control groups, respectively (the MAF in the European population is 11%). The MAF frequency of rs1548357 was 30.4% and 31.4%, respectively (the MAF in the European population is 30%). MAF of rs7212 was 3.6% and 2.9%, respectively (the MAF in the European population is 4.5%) [17].

Additionally, we explored the association between our selected polymorphisms and glucose levels, HbA1c, arterial hypertension, statin use, and metformin use. We found no statistically significant associations between the selected variables and our studied polymorphisms (Appendix A).

## 3. Discussion

The aims of our study were to investigate the association between the rs8140110 polymorphism in the *TXN2* gene, the rs1548357 polymorphism in the *TXNRD2* gene, and the rs7212 polymorphism in the *TXNIP* gene with DN in Slovenian patients with T2DM. Our study is the first to investigate the association between these polymorphisms and DN in patients with T2DM.

We discovered that patients with the T allele of the rs8140110 polymorphism were less likely to have DN in comparison to patients with the C allele (95% OR: 0.52 (0.36–0.74); *p* < 0.001). Patients with the TT genotype were almost five times less likely to have DN in comparison to patients with the CC genotype (95% OR: 0.21 (0.03–0.84); *p* = 0.0498). The same protective effect of the rs8140110 polymorphism was found when comparing the TC genotype with the CC genotype (95% OR: 0.54 (0.36–0.81); *p* = 0.0034). We also constructed a dominant and recessive model of inheritance. The dominant inheritance model showed that patients with the TT or TC genotypes were almost half as likely to have DN in comparison to patients with the CC genotype (95% OR: 0.51 (0.34–0.75); *p* < 0.001). In our logistic regression analysis, we adjusted for the duration of T2DM, duration of hypertension, systolic blood pressure, waist circumference, DR, diabetic neuropathy, HbA1c, fasting glucose, urea, creatinine, cystatin C, and urine albumin–creatinine ratio. Poor glycemic control, a longer duration of T2DM, and hypertension are well-established risk factors for DN [18]. However, we did not find an association between the rs1548357 or rs7212 polymorphisms and DN. The distribution of alleles and genotypes did not differ significantly between the case group and the control group (*p* > 0.05 in both cases). Patients in the case group had a longer duration of T2DM and hypertension, higher systolic blood pressure, and larger waist circumference. They also had higher levels of fasting glucose and HbA1c, as well as a higher prevalence of other complications related to diabetes (DR and diabetic neuropathy).

At the time of writing this article, we did not find any published articles that investigated the role of *TXN2* polymorphisms and DN. Several authors have investigated polymorphisms in the *TXNRD2* and *TXNIP* and DN, as well as other complications related to diabetes. Kariž et al. reported an association between the rs1548357 polymorphism of *TXNRD2* and myocardial infarction in patients with T2DM. They discovered that the C allele of the rs1548357 polymorphism could have a protective role against macrovascular complications of T2DM (atherosclerosis) in Slovenian patients [19]. In another study, Mankoč Ramuš et al. investigated the association between several polymorphisms in *TXN2*, *TXNRD2*, and *TXNIP* and DR. They did not find an association between rs8140110, rs1548357, or rs7212 and DR, but they did find an association between rs4485648 of the *TXNRD2* gene and DR [20]. These two studies were also conducted on a Slovenian population and from these results we can infer that genetic variation in the thioredoxin antioxidant system influences the development of complications related to diabetes in a Slovenian population.

Three other polymorphisms in *TXNRD2* were found to be associated with diabetic kidney disease in Greek patients with T2DM. Roumeliotis et al. performed a study that investigated the association between genes that are involved in oxidative stress and DKD. The polymorphisms rs737866, rs12106549, and rs201971987 in the *TXNRD2* gene were found to be associated with DKD. In contrast with our results, they did not identify rs8140110 as being associated with DN. The key differences between our study and the study by Roumeliotis et al. are the number of participants and the population. They included only 121 patients with T2DM and DKD and 220 T2DM patients without DKD, so their sample size was smaller than ours. Also, they conducted their study on a Greek population which could explain the different conclusions [21].

Ferreira et al. investigated several polymorphisms in the *TXNIP* gene and reported an association of the rs7211 C/rs7212 G haplotype with diabetes in a Brazilian population [22]. Alvim et al. also reported an association between rs7212 and arterial stiffness (determined by pulse-wave velocity) in Brazilians with T2DM [23]. Liang et al. investigated the association between *TXNIP* rs7212 and T2DM in a Chinese Han population. They discovered an association between the G allele and susceptibility to T2DM, but their sample size was quite small (161 patients and 146 controls) [24].

Several studies have reported the increased expression of TXNIP in kidney samples of patients with DN [25,26,27]. Advani et al. reported increased TXNIP expression in rat models of diabetes, as well as samples from human patients with DN. They also investigated whether the expression of TXN was affected by diabetes, but they did not find a difference in the TXN expression in diabetes [25]. The thioredoxin antioxidant system’s crucial role in DN’s pathogenesis was also demonstrated in animal experiments. TXNIP knockout mice were protected from renal fibrosis, the accumulation of extracellular matrix, decreases in renal function, and albuminuria after diabetes was induced [26].

An important limitation of our preliminary study is the small sample size in combination with the relatively low MAF of the studied polymorphisms. Only a few homozygotes of the *TXN2* rs8140110 and *TXNIP* rs7212 MAF alleles were present in our preliminary study. Studies with more participants are needed to verify our results. We also did not take into account the influence of other polymorphisms in the *TXN2*, *TXNIP*, and *TXNRD2* genes or other genes that may have influenced our results. Our study was a retrospective one, a study of a prospective nature would be more valuable. However, we have partially addressed this by including only patients who had had T2DM for at least 10 years. Studies that would investigate the expression of TXN2 in the kidneys of patients with DN and controls would complement the results of our study. Other populations should also be studied, as well as other polymorphisms in the thioredoxin system.

To conclude, the rs8140110 polymorphism in the *TXN2* gene is associated with DN in Slovenian patients with T2DM. The TT and TC genotypes are significantly less likely to develop DN in comparison to the CC genotype according to the dominant model of inheritance. Because of the intronic location the rs8140110 and the limitations of our study stated above, we cannot determine the exact mechanism by which it is associated with DN. It could be involved in the regulation of the expression of *TXN* or its posttranscriptional modification, but more research is needed. The thioredoxin antioxidant system is an interesting target for future research and also a potential therapeutic target.

## 4. Materials and Methods

### 4.1. Patients

We included 897 unrelated Slovene patients in our retrospective cross-section association study. All of the patients were Caucasian and had had T2DM for at least 10 years. In total, 553 patients without DN were included in the control group and 344 patients with DN were included in the case group. The diagnosis of T2DM and DN was made using the World Health Organization (WHO) criteria. The exclusion criteria for our study were significant heart failure (New York Heart Association (NYHA) Classification II–IV), active infection, overt nephropathy, poor glycemic control (HbA1c greater than 10%), overt nephropathy, and the presence of other causes of renal disease. The flowchart containing the inclusion and exclusion criteria can be found below (Figure 1). Information regarding the participants’ age, sex, smoking status, systolic and diastolic blood pressure, duration of arterial hypertension, history of carotid artery disease, duration of T2DM, and the presence of other microvascular T2DM complications (presence and duration of DR, diabetic neuropathy, and diabetic foot) were obtained with the help of a questionnaire and the patients’ medical records. The mass, height, and waist circumference of the patients were measured. Peripheral venous blood samples were taken after an overnight fast.

All the participants gave their informed consent to be included in our study. We performed our study in accordance with the Helsinki declaration. The Slovenian medical ethics committee approved our study: the ethics approval numbers for our study are 105/12/2011 and 0120-163/2024.

### 4.2. Biochemical Analysis

Fasting glucose, HbA1c, hemoglobin, total cholesterol, HDL-cholesterol, LDL-cholesterol, triglycerides, cystatin C, creatinine, and urea were measured using standard biochemical analysis. The albumin–creatinine ratio was determined in three different urine samples. The MDRD study equation and cystatin C were used to obtain the estimated glomerular filtration rate (eGFR).

### 4.3. Genotyping

We used the QIAamp DNA Blood Mini Kit (Qiagen GmbH, Hilden, Germany) to extract the genomic DNA from leukocytes in 100 μL of peripheral venous blood. We analyzed the TXN2 rs8140110, TxnRD2 rs1548357, and TXNIP rs7212 polymorphisms using the StepOne™ (48-well) Real-Time PCR Systems (Applied Biosystem by Life Technologies, Waltham, MA, USA) and KBioscience Ltd. (LGC, Teddington, UK) competitive allele-specific fluorescence-based PCR (KASPar) assay. FAM and HEX were the fluorophores that were used to differentiate between the alleles. The sequence of the TXN2 rs8140110 assay used was as follows: CTTCCCCTTCCTCTCAATTTTGCCA[C/T]GAACTAAAACTGCCCTAAAAACAT; the sequence of the TxnRD2 rs1548357 assay used was as follows: GCTGAGCAAAGACAGTTGCCTGCCG[T/C]GTACATGCCAGCAAACATGCCTGG; the sequence of the TXNIP rs7212 assay used was as follows: ATTTGGAGGTTCTGATCACAGGGTT[G/C]GGCATCTTGATCAAGAGTTCCTCCT. The total volume of the reaction was 6 µL, and the volume of DNA was 0.6 µL. We used the “two-step touchdown” amplification protocol with 36 cycles of amplification and an additional 10 cycles of recycling. Additional information is available at https://www.biosearchtech.com/support/faqs/kasp-genotyping-assays/, accessed on 27 December 2024.

### 4.4. Statistical Analysis

The Shapiro–Wilk test was used to determine the normality of distribution. We used Mann–Whitney’s U test or Student’s *t*-test to compare continuous variables. To compare discrete variables, we used the Chi-square test. Logistic regression after adjusting for waist circumference, systolic blood pressure, S-fasting glucose, triglycerides, urea, diabetic retinopathy, and diabetic neuropathy was used to analyze the relationship between TXN2 rs8140110, TxnRD2 rs1548357, and TXNIP rs7212 and DN. A value of *p* < 0.05 was considered statistically significant. We used the Chi-square goodness-of-fit test to determine the deviation for the Hardy–Weinberg equilibrium (HWE) (https://www.socscistatistics.com/tests/goodnessoffit/default2.aspx, accessed on 18 November 2024).

## Figures and Tables

**Figure 1 ijms-26-01832-f001:**
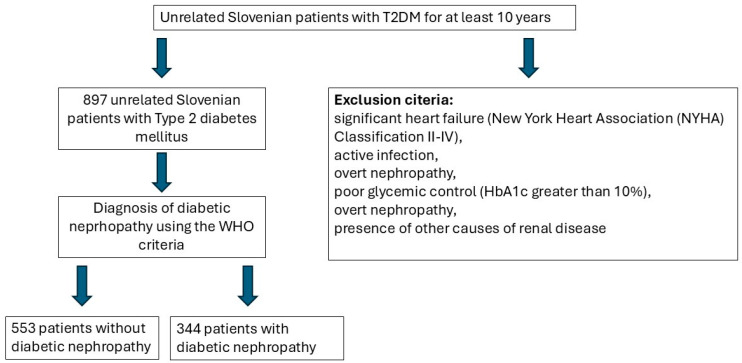
Flowchart of the inclusion and exclusion criteria for our study.

**Table 1 ijms-26-01832-t001:** Clinical and laboratory characteristics of cases and controls.

	Case (N = 344)	Control (N = 553)	*p* Value
**Sex [M]**	201 (58.4%)	318 (57.5%)	0.78
**Age [years] ^a^**	65.3 ± 9.4	65.4 ± 8.6	0.90
**Duration of T2DM [years] ^b^**	16.0 [13.0–20.0]	15.0 [12.0–19.0]	**<0.001**
**Duration of hypertension [years] ^b^**	12.0 [7.0–18.0]	11.0 [5.0–16.0]	**0.0058**
**SBP [mmHg] ^b^**	155.0 [142.7–167.0]	145.0 [135.0–160.0]	**<0.001**
**DBP [mmHg] ^b^**	85.0 [77.0–90.0]	80.0 [75.0–90.0]	0.0759
**BMI ^a^**	30.39 ± 3.56	29.94 ± 3.91	0.08
**Waist circumference ^a^**	108.56 ± 10.41	104.59 ± 11.78	**<0.001**
**Active smokers**	32 (9.3%)	70 (12.7%)	0.12
**CVD**	89 (25.9%)	165 (29.8%)	0.20
**DR**	149 (43.3%)	115 (20.8%)	**<0.001**
**Duration of DR [years] ^b^**	5.0 [5.0–6.0]	6.0 [4.0–8.0]	**<0.001**
**Diabetic neuropathy**	69 (20.1%)	39 (7.1%)	**<0.001**
**S-HbA1c [%] ^b^**	7.70 [6.90–8.72]	7.50 [6.90–8.20]	**<0.001**
**S-fasting glucose [mmol/L] ^b^**	8.70 [7.18–10.33]	8.10 [6.80–9.50]	**<0.001**
**S-Hb [g/L] ^a^**	138.87 ± 13.26	139.65 ± 11.84	0.36
**S-urea [mmol/L] ^b^**	6.20 [5.20–7.70]	5.90 [4.80–7.40]	**0.0049**
**S-creatinine [μmol/L] ^b^**	80.0 [68.0–102.0]	77.0 [65.0–90.0]	**0.0025**
**Male sex ^b^**	89.0 [73.0–106.0]	84.0 [71.0–98.0]	**0.0025**
**Female sex ^b^**	72.0 [58.0–88.0]	70.0 [61.0–80.0]	0.097
**eGFR [MDRDequation, mL/min] ^b^**	60.0 [58.0–64.0]	77.0 [60.0–88.0]	**<0.001**
**Male sex ^b^**	60.0 [60.0–60.0]	79.0 [68.0–87.0]	**<0.001**
**Female sex ^b^**	62.0 [59.0–66.0]	60.0 [60.0–60.0]	**<0.001**
**S-cystatin C [mg/L] ^b^**	0.82 [0.69–1.03]	0.74 [0.65–0.86]	**<0.001**
**S-Total cholesterol [mmol/L] ^b^**	4.35 [3.80–5.20]	4.40 [3.90–5.10]	0.82
**S-HDL [mmol/L] ^b^**	1.20 [1.00–1.40]	1.20 [1.00–1.40]	0.86
**S-LDL [mmol/L] ^b^**	2.40 [2.00–3.00]	2.40 [2.00–3.00]	0.66
**S-TGS [mmol/L] ^b^**	1.60 [1.10–2.32]	1.40 [1.00–2.10]	**0.0123**
**U-albumin–creatinine ratio [g/mol], sample no. 1 ^b^**	7.92 [3.51–22.67]	1.00 [0.60–1.58]	**<0.001**
**U-albumin–creatinine ratio [g/mol], sample no. 2 ^b^**	7.97 [3.58–24.51]	1.03 [0.68–1.70]	**<0.001**
**U-albumin–creatinine ratio [g/mol], sample no. 3 ^b^**	8.18 [3.52–22.82]	1.02 [0.68–1.70]	**<0.001**
**Statin use**	275 (79.9%)	423 (76.5%)	0.18
**Metformin use**	111 (32.3%)	208 (37.6%)	0.10

Table legend: ^a^ Student’s *t*-test was used to compare the groups; ^b^ Mann–Whitney’s U test was used to compare the groups. The value of *p* < 0.05 was considered statistically significant. The Shapiro–Wilk test was used to determine the normality of distribution. Statistically significant values of *p* are written in bold. Abbreviations: T2DM—type 2 diabetes mellitus; SBP—systolic blood pressure; DBP—diastolic blood pressure; BMI—body mass index; CVD—cardiovascular disease; DR—diabetic retinopathy; Hb—hemoglobin; eGFR—estimated glomerular filtration rate; HDL—high-density lipoprotein cholesterol; LDL—low-density lipoprotein cholesterol; TGS—triglycerides.

**Table 2 ijms-26-01832-t002:** Distribution of rs8140110 genotypes, alleles, and dominant/recessive models of inheritance (logistic regression analysis was used).

TXN2_rs8140110	Case (N = 344)	Control (N = 553)	*p* Value	Adjusted OR (95% CI)
**TT**	2 (0.6%)	12 (2.2%)	**0.0322**	**0.21 (0.03–0.84) [*p* value: 0.0498]**
**TC**	51 (14.8%)	107 (19.3%)		**0.54 (0.36–0.81) [*p* value: 0.0034]**
**CC**	291 (84.6%)	434 (78.5%)		ref.
**ALLELES**				
**T (MAF)**	55 (8.0%)	131 (11.8%)	**0.0093**	**0.52 (0.36–0.74) [*p* value: <0.001]**
**C**	633 (92.0%)	975 (88.2%)		ref.
**HWE (*p* value)**	0.88	0.0841		
**DOMINANT**				
**TT + TC**	53 (15.4%)	119 (21.5%)	**0.0238**	**0.51 (0.34–0.75) [*p* value: <0.001]**
**CC**	291 (84.6%)	434 (78.5%)		ref.
**RECESSIVE**				
**TT**	2 (0.6%)	12 (2.2%)	0.0620	0.23 (0.03–0.92) [*p* value: 0.0661]
**TC + CC**	342 (99.4%)	541 (97.8%)		ref.

Logistic regression analysis was adjusted for the duration of type 2 diabetes mellitus, duration of hypertension, systolic blood pressure, waist circumference, diabetic retinopathy, diabetic neuropathy, HbA1c, fasting glucose, urea, creatinine, cystatin C, and urine albumin–creatinine ratio. The value of *p* < 0.05 was considered statistically significant. Statistically significant values are written in bold. Abbreviations: MAF—minor allele frequency; HWE—Hardy–Weinberg equilibrium; OR—odds ratio; CI—confidence interval.

**Table 3 ijms-26-01832-t003:** Distribution of rs1548357 genotypes, alleles, and dominant/recessive models of inheritance (logistic regression analysis was used).

TXNRD2_rs1548357	Case (N = 344)	Control (N = 553)	*p* Value	Adj OR (95% CI)
**CC**	31 (9.0%)	53 (9.6%)	0.90	0.85 (0.5–1.44) [*p* value: 0.56]
**CT**	147 (42.7%)	241 (43.6%)		0.85 (0.62–1.17) [*p* value: 0.32]
**TT**	166 (48.3%)	259 (46.8%)		ref.
**ALLELES**				
**C (MAF)**	209 (30.4%)	347 (31.4%)	0.66	0.9 (0.72–1.13) [*p* value: 0.35]
**T**	479 (69.6%)	759 (68.6%)		ref.
**HWE (*p* value)**	0.85	0.78		
**DOMINANT**				
**CC + CT**	178 (51.7%)	294 (53.2%)	0.68	0.85 (0.63–1.15) [*p* value: 0.30]
**TT**	166 (48.3%)	259 (46.8%)		ref.
**RECESSIVE**				
**CC**	31 (9.0%)	53 (9.6%)	0.77	0.92 (0.55–1.52) [*p* value: 0.75]
**CT + TT**	313 (91.0%)	500 (90.4%)		ref.

Logistic regression analysis was adjusted for the duration of type 2 diabetes mellitus, duration of hypertension, systolic blood pressure, waist circumference, diabetic retinopathy, diabetic neuropathy, HbA1c, fasting glucose, urea, creatinine, cystatin C, and urine albumin–creatinine ratio. The value of *p* < 0.05 was considered statistically significant. Abbreviations: MAF—minor allele frequency; HWE—Hardy–Weinberg equilibrium; OR—odds ratio; CI—confidence interval.

**Table 4 ijms-26-01832-t004:** Distribution of rs7212 genotypes, alleles, and dominant/recessive models of inheritance (logistic regression analysis was used).

TXNIP_rs7212	Case (N = 344)	Control (N = 553)	*p* Value	Adj OR (95% CI)
**CC**	1 (0.3%)	1 (0.2%)	0.69	0.96 (0.04–25.1) [*p* value: 0.98]
**CG**	23 (6.7%)	30 (5.4%)		1.1 (0.59–1.99) [*p* value: 0.77]
**GG**	320 (93.0%)	522 (94.4%)		ref.
**ALLELES**				
**C (MAF)**	25 (3.6%)	32 (2.9%)	0.38	1.08 (0.61–1.9) [*p* value: 0.79]
**G**	663 (96.4%)	1074 (97.1%)		ref.
**HWE (*p* value)**	0.40	0.42		
**DOMINANT**				
**CC + CG**	24 (7.0%)	31 (5.6%)	0.41	1.09 (0.6–1.96) [*p* value: 0.78]
**GG**	320 (93.0%)	522 (94.4%)		ref.
**RECESSIVE**				
**CC**	1 (0.3%)	1 (0.2%)	0.73	0.95 (0.04–24.98) [*p* value: 0.97]
**CG + GG**	343 (99.7%)	552 (99.8%)		ref.

Logistic regression analysis was adjusted for the duration of type 2 diabetes mellitus, duration of hypertension, systolic blood pressure, waist circumference, diabetic retinopathy, diabetic neuropathy, HbA1c, fasting glucose, urea, creatinine, cystatin C, and urine albumin–creatinine ratio. The value of *p* < 0.05 was considered statistically significant. Abbreviations: MAF—minor allele frequency; HWE—Hardy–Weinberg equilibrium; OR—odds ratio; CI—confidence interval.

## Data Availability

The data presented in this study are available on request from the corresponding author due to sensitive information (patients’ clinical data).

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
