# Peer review of "Single-Nucleotide Polymorphisms in the Thioredoxin Antioxidant System and Their Association with Diabetic Nephropathy in Slovenian Patients with Type 2 Diabetes—A Preliminary Study"

_ijms, 2025, doi:10.3390/ijms26051832_

Round 1
Reviewer 1 Report
Comments and Suggestions for Authors
The manuscript entitled “Single nucleotide polymorphisms in the thioredoxin antioxidant system and their association with diabetic nephropathy in Slovenian patients with type 2 diabetes” by Jernej Letonja et al, presents an original research article within the scope of the journal, and addresses an interesting and important topic with high interest for the biomedical, and medical community.
The authors investigated the association of polymorphisms in genes for thioredoxin (Txn2), thioredoxin reductase (TxnRD2), and thioredoxin interacting protein (TXNIP), components of the thioredoxin antioxidant system, with the diabetic nephropathy.
Overall, this is a very interesting study. The manuscript was well prepared, and the limitations of the study are clearly stressed. The experimental design is adequate, experiments have been carefully conducted, the analyses are appropriate, the findings are promising and the conclusions are supported by the data. The literature search is adequate and up to date. Nonetheless, some points should be amended:
11. Due to the relatively small number of patients, my suggestion is to modify the title by emphasizing that this is a preliminary or pilot study, and making appropriate changes throughout the text of the manuscript.
- The manuscript would benefit from a Flowchart showing inclusion/exclusion criteria, classification, and number of patients.
- Please check the p value in line 99.
- What are the future possibilities that could be undertaken as a result of the findings of this study?
- Please change “body weight” to “body mass” throughout the manuscript.
- All abbreviations need to be explained at first use in the text (e.g. IL, DKD...), Abstract and Table/Figure legends, and when introduced the abbreviation should be exclusively used throughout the whole text (e.g. ASK1). My suggestion is to include all abbreviations in the Abbreviation list.
Author Response
The manuscript entitled “Single nucleotide polymorphisms in the thioredoxin antioxidant system and their association with diabetic nephropathy in Slovenian patients with type 2 diabetes” by Jernej Letonja et al, presents an original research article within the scope of the journal, and addresses an interesting and important topic with high interest for the biomedical, and medical community.
The authors investigated the association of polymorphisms in genes for thioredoxin (Txn2), thioredoxin reductase (TxnRD2), and thioredoxin interacting protein (TXNIP), components of the thioredoxin antioxidant system, with the diabetic nephropathy.
Overall, this is a very interesting study. The manuscript was well prepared, and the limitations of the study are clearly stressed. The experimental design is adequate, experiments have been carefully conducted, the analyses are appropriate, the findings are promising and the conclusions are supported by the data. The literature search is adequate and up to date. Nonetheless, some points should be amended:
Thank you for your kind and insightful comments.
- Due to the relatively small number of patients, my suggestion is to modify the title by emphasizing that this is a preliminary or pilot study, and making appropriate changes throughout the text of the manuscript.
We have made the appropriate changes to the title and the manuscript.
- The manuscript would benefit from a Flowchart showing inclusion/exclusion criteria, classification, and number of patients.
We have included a Flowchart in the Materials & methods section
- Please check the p value in line 99.
We have corrected the p value in line 99.
- What are the future possibilities that could be undertaken as a result of the findings of this study?
We have included a short paragraph about the future research possibilities in the discussion:
»Studies that would investigate the expression of Txn2 in kidneys of patients with DN and controls would complement the results of our study. Other populations should also be studied as well as other polymorphisms in the thioredoxin system.«
- Please change “body weight” to “body mass” throughout the manuscript.
We have changed body weight to body mass throughout the manuscript.
- All abbreviations need to be explained at first use in the text (e.g. IL, DKD...), Abstract and Table/Figure legends, and when introduced the abbreviation should be exclusively used throughout the whole text (e.g. ASK1). My suggestion is to include all abbreviations in the Abbreviation list
Thank you for noticing this. We have made sure all abbreviations are included in the Abbreviation list and that only abbreviations are exclusively used once introduced in the manuscript.
Reviewer 2 Report
Comments and Suggestions for Authors
The article by Jernej Letonja and Petra Nussdorfer has an interesting idea. However, I think that the article in its current form is not up to the IJMS level. However, due to the topic, I think that the quality of the article could be improved with new results. I hope the authors find this feedback constructive and helpful. Therefore, please consider the following analyses:
1. A lot of laboratory data was collected, but their statistical analysis was not carried out. For example, glucose has a significant role in the functioning of the thioredoxin system. In the discussion, they explain the effect of each SNP on the cardiovascular system. Despite this, the relationship between laboratory parameters and genetic background was not examined.
2. It would be good to mention in the article which is the wild-type allele for each SNP, what is the literature MAF, and how the MAF value of the groups examined here changes compared to it.
3. It should be specifically described in the titles of the tables what statistics were used, with what p value...
4. Information on medication intake would be important. Metformin is excreted through the kidneys, SGLT2 inhibitors also affect the kidneys... We should see if there was a difference between the two groups.
5. Since these polymorphisms have several cardiological implications, the patients' statin use and the relationship of the genotypes with the drugs would also be important information.
6. It should be better explained why, in your opinion, the more common allele increases in kidney patients? Shouldn't it be better to turn things around and say that the less common allele is protective against the disease?
7. Is there information on what other comorbidities the patients have?
8. I find it interesting that the albumin creatinine ratio shows an increasing trend in the patients at the 3 sampling points. Could you explain why? At what times was the sampling done?
9. Tables 2, 3, 4 are difficult to interpret, please think about a more transparent design.
Author Response
The article by Jernej Letonja and Petra Nussdorfer has an interesting idea. However, I think that the article in its current form is not up to the IJMS level. However, due to the topic, I think that the quality of the article could be improved with new results. I hope the authors find this feedback constructive and helpful. Therefore, please consider the following analyses:
Thank you for your helpful and insightful comments.
- A lot of laboratory data was collected, but their statistical analysis was not carried out. For example, glucose has a significant role in the functioning of the thioredoxin system. In the discussion, they explain the effect of each SNP on the cardiovascular system. Despite this, the relationship between laboratory parameters and genetic background was not examined.
Thank you for this insightful comment. We have examined the relationship between glucose and HbA1C and the studied polymorphisms. We have included the data in Supplementary tables.
- It would be good to mention in the article which is the wild-type allele for each SNP, what is the literature MAF, and how the MAF value of the groups examined here changes compared to it.
Thank you for this comment. We have included the information in the manuscript.
- It should be specifically described in the titles of the tables what statistics were used, with what p value...
We have included this information under the Tables. We believe that including all this information in the title of the Table would make the titles too long, yet the information is still easily available for the reader.
- Information on medication intake would be important. Metformin is excreted through the kidneys, SGLT2 inhibitors also affect the kidneys... We should see if there was a difference between the two groups.
Thank you for this insightful comment. Our study is a retrospective one and we have gathered our participans over the course of several years. SGLT2 inhibitors were not yet available at the beginning of the study and were only starting to be introduced in regular practise towards the end of our participant gathering phase. So no participants in our study were perscribed SGLT2 inhibitors (at the time of the data & blood sample gathering). We have information regarding metformin use, and have included it in the manuscript.
- Since these polymorphisms have several cardiological implications, the patients' statin use and the relationship of the genotypes with the drugs would also be important information.
We have included this information in our manuscript. There was no difference between the groups regarding statin use.
- It should be better explained why, in your opinion, the more common allele increases in kidney patients? Shouldn't it be better to turn things around and say that the less common allele is protective against the disease?
Thank you for this question. We chose to present our findings the way we did because the T allele of the rs8140110 is the ancestral allele.
- Is there information on what other comorbidities the patients have?
We have gathered information regarding microvascular T2DM complications (presence and duration of DR, diabetic neuropathy, diabetic foot) and arterial hypertension. Sadly, we did not gather information about all the other diagnoses of the patients.
- I find it interesting that the albumin creatinine ratio shows an increasing trend in the patients at the 3 sampling points. Could you explain why? At what times was the sampling done?
Thank you for this comment. Urine samples were gather in the morning (consecutive days). This ratio was indeed higher for sample 3 than sample 1, however the difference was minimal. We did not have any idea for the reason of that.
- Tables 2, 3, 4 are difficult to interpret, please think about a more transparent design
Thank you for this comment. We have attempted to reform the tables, but we could not find a design that would be equally effective at conveying our results and at the same time not splitting the results into several different tables.
Round 2
Reviewer 2 Report
Comments and Suggestions for Authors
The article has improved significantly; thank you for considering my suggestions. However, I still have some issues with the article.
1. The tables need to be standardized (italics, highlights, centering, and proper description of test names in figure captions).
2. The methodology section needs to be more specific about the genotyping method. Provide the SNP assay ID number or describe the specific sequences used, including dyes.
3. Why was the "two-step touchdown" PCR needed? Why did the traditional TaqMan PCR setup not work? How did this affect sensitivity? Were other stabilizing factors, such as betaine, needed?
4. The statistical analysis should describe the normality test to determine whether a parametric or nonparametric statistical test is appropriate for the sample analysis.
5. Do not use asterisks for p-values in tables, as this is the traditional symbol for statistical significance, which can be misleading.
6. In Table 1, indicate where parametric and nonparametric tests were used.
7. The p-value should be written as a lowercase "p".
8. Make sure to explain all abbreviations used in the tables, no matter how trivial they may seem.
9. Please describe why you chose these SNPs for your study. What criteria were used to select them?
10. In the discussion, please consider how an intronic mutation could cause such changes.
Author Response
The article has improved significantly; thank you for considering my suggestions. However, I still have some issues with the article.
- The tables need to be standardized (italics, highlights, centering, and proper description of test names in figure captions).
- Do not use asterisks for p-values in tables, as this is the traditional symbol for statistical significance, which can be misleading.
In Table 1, indicate where parametric and nonparametric tests were used.
The p-value should be written as a lowercase "p".
Make sure to explain all abbreviations used in the tables, no matter how trivial they may seem.
Thank you for your comments. We have made the appropriate adjustments to the Tables as per your request.
The methodology section needs to be more specific about the genotyping method. Provide the SNP assay ID number or describe the specific sequences used, including dyes.
We have provided the sequences of the assays as well as the dyes in the materials & methods section.
Why was the "two-step touchdown" PCR needed? Why did the traditional TaqMan PCR setup not work? How did this affect sensitivity? Were other stabilizing factors, such as betaine, needed?
We used the »two-step touchdown« protocol (as described in the article) as we have discovered it to be the most optimal protocol for minimizing the total volume of the reaction without sacrificing the sensitivity. Larger total volumes of reaction are used in the beginning to obtain positive controls as well as throughout the experiment to double-check our results obtained by lower total volumes. We did not need to add other stabilizing factors to the reaction. Only the stated chemical were used.
The statistical analysis should describe the normality test to determine whether a parametric or nonparametric statistical test is appropriate for the sample analysis.
Thank you for this comment. We have added the information in the statistical analysis section.
Please describe why you chose these SNPs for your study. What criteria were used to select them?
We selected the SNPs based on a literature review and intriguing findings in some other populations.
In the discussion, please consider how an intronic mutation could cause such changes
Thank you for these comments. We have strenghtened the discussion section (marked in yellow): »Because of the intronic location the rs8140110 and the limitations of our study stated above we cannot determine the exact mechanism by which it is associated with DN. It could be involved in the regulation of the expression of TXN or its posttranscriptional modification, but more research is needed«.
Round 3
Reviewer 2 Report
Comments and Suggestions for Authors
All my questions were answered.